# Can Microfluidics Improve Sperm Quality? A Prospective Functional Study

**DOI:** 10.3390/biomedicines12051131

**Published:** 2024-05-20

**Authors:** Fernando Meseguer, Carla Giménez Rodríguez, Rocío Rivera Egea, Laura Carrión Sisternas, Jose A. Remohí, Marcos Meseguer

**Affiliations:** 1IVIRMA Global Research Alliance, IVIRMA Valencia, Plaza de la Policía Local 3, 46015 Valencia, Spain; rocio.rivera@ivirma.com (R.R.E.); marcos.meseguer@ivirma.com (M.M.); 2IVIRMA Global Research Alliance, IVI Foundation, Instituto de Investigación Sanitaria La Fe (IIS La Fe), 46026 Valencia, Spain; carla.gimenez@ivirma.com (C.G.R.); laura.carrion@ivirma.com (L.C.S.)

**Keywords:** microfluidics, sperm selection, sperm quality, swim-up, density gradient

## Abstract

The same sperm selection techniques in assisted reproduction clinics have remained largely unchanged despite their weaknesses. Recently, microfluidic devices have emerged as a novel methodology that facilitates the sperm selection process with promising results. A prospective case-control study was conducted in two phases: 100 samples were used to compare the microfluidic device with Density Gradient, and another 100 samples were used to compare the device with the Swim-up. In the initial phase, a significant enhancement in progressive motility, total progressive motile sperm count, vitality, morphology, and sperm DNA fragmentation were obtained for the microfluidic group compared to Density Gradient. Nevertheless, no statistically significant differences were observed in sperm concentration and chromatin structure stability. In the subsequent phase, the microfluidic group exhibited significant increases in sperm concentration, total progressive motile sperm count, and vitality compared to Swim-up. However, non-significant differences were seen for progressive motility, morphology, DNA structure stability, and DNA fragmentation. Similar trends were observed when results were stratified into quartiles. In conclusion, in a comparison of microfluidics with standard techniques, an improvement in sperm quality parameters was observed for the microfluidic group. However, this improvement was not significant for all parameters.

## 1. Introduction

The World Health Organization (WHO) recognizes infertility as a disease affecting a significant global population of approximately 186 million individuals [1,2]. Among infertility cases, the male factor contributes to 50% of them, being responsible for the inability to achieve pregnancy in 20% to 30% of cases [3,4]. Assisted reproductive techniques (ART) have emerged as effective interventions for couples facing infertility [5]. The primary ART procedures currently available include intrauterine insemination (IUI), in vitro fertilization (IVF), and intracytoplasmic sperm injection (ICSI). Regardless of the specific technique employed, prior preparation of the semen sample is essential to mimic the intricate selection process of optimal spermatozoa that naturally occurs within the female reproductive tract. This preparation aims to effectively identify and isolate motile and morphologically normal spermatozoa from other cellular components and potentially harmful substances [6].

Currently, the laboratory techniques commonly employed for sperm selection in assisted reproduction treatments include Swim-up and Density Gradient Centrifugation (DGC). Both techniques involve a centrifugation step. This process leads to an elevation of reactive oxygen species (ROS) levels, consequently causing DNA fragmentation [7,8]. Sperm DNA damage plays a crucial role in determining sperm quality and has been associated with lower pregnancy rates, impaired preimplantation embryo development, early pregnancy loss, and an increased risk of disease in offspring [8]. Sperm lack inherent DNA repair mechanisms and are unable to repair DNA breaks after spermatogenesis. Nevertheless, the oocyte possesses some DNA repair mechanisms, which can repair damage to the sperm’s DNA, thereby allowing for successful fertilization [9,10,11,12]. However, the outcome relies on the level of genetic damage, as high fragmentation or poor-quality oocytes compromise the achievement of fertilization and subsequent embryo development [9,13,14,15,16,17].

Nowadays, microfluidics is gaining significant interest as a method for sperm selection. Over the past two decades, microfluidics has shown great potential in various assisted reproduction procedures, including semen analysis for male infertility diagnosis [18,19], sperm selection [20,21], oocyte analysis and denudation [22], insemination [23,24], embryo culture [25,26], and cryopreservation [27,28]. Despite the successful results observed, the integration of microfluidics into clinical practice is still in its early stages of development [29]. In the field of sperm selection, the microfluidic device emerges as a promising method that offers efficiency, simplicity, and high selectivity for selecting viable sperm. This innovative approach operates by directing sperm through confined microchannels with diverse geometries, effectively mimicking the natural sperm pathway and the process of in vivo selection [30]. One notable advantage of this device based on microfluidics is that it eliminates the need for prior centrifugation steps, mitigating the production of ROS, which can potentially harm sperm DNA [31,32]. Furthermore, the microfluidic device overcomes the limitations associated with traditional methods, such as reliance on bulky and costly equipment, prolonged waiting times, and the requirement for highly skilled technicians [33].

Considering these merits, the microfluidic device hold tremendous potential to become a routine methodology for clinical sperm selection. However, there are limited studies available that demonstrate its performance in comparison to conventional techniques. Therefore, the objective of this study is to compare sperm quality parameters between the novel microfluidic device and conventional techniques (DGC and Swim-up).

## 2. Materials and Methods

### 2.1. Study Design and Population

The present research work is a prospective study carried out at IVI Valencia, aiming to compare the efficacy of the novel microfluidic device called SwimCount^TM^ Harvester (MotilityCount ApS, Copenhagen, Denmark) with the conventional methodologies (Density Gradient Centrifugation and Swim-up) employed in andrology laboratories for sperm selection. The study was approved by our institutional research board and ethical committee with a code “1902-VLC-022-MM”. The study was conducted in two different phases: the first phase involved comparing the SwimCount^TM^ Harvester with DGC, while the second phase was focused on comparing the microfluidic device with the Swim-up technique. A total of 200 males, with sperm samples with a concentration of more than one million per milliliter and more than 1% progressive sperm, were recruited for the study for two consecutive years. All participants provided their informed consent voluntarily by signing the appropriate documentation. For each participant, a semen sample was collected, which was subsequently divided into two equal volumes. One volume underwent sperm selection using the conventional methodology routinely employed at the clinic, while the other volume was processed using the microfluidic device (Figure 1).

### 2.2. Semen Processing and Analysis

One of the methodologies employed for semen processing entailed the utilization of the microfluidic device, SwimCount^TM^ Harvester (MotilityCount ApS, Copenhagen, Denmark). TOPAS^®^ COC (cyclic olefin copolymer) is used for the production of the microfluidic device, which consists of two chambers divided by a membrane with 10 μm microchannels, enabling precise fluid management within the device. A volume of 1 mL of sperm sample was introduced into the lower chamber, followed by the addition of 0.8 mL of Sequential FertTM (Origio, Måløv, Denmark), into the upper chamber. It was then incubated for 30 min at 37 °C. Upon the expiration of the duration, a volume of 0.8 mL of Sequential FertTM, wherein the most optimal sperm population resides, was extracted using a syringe. Only the best quality spermatozoa, which due to their intrinsic properties could actively swim and overcome the membrane, reached the upper chamber, where they were subsequently collected.

Another methodology used to process the semen was DGC. This technique separated the spermatozoa according to their density. Three phases are distinguished from lower to higher density, with a percentage of colloidal silica of 45, 70, and 95% in each phase (Sil-Select stock^TM^ with gentamicin, FertiPro, Beernem, Belgium). The semen sample was deposited on top of these phases and centrifuged at 432× *g* for 11 min. The pellet was collected and resuspended in a FertiCult^TM^ Flushing Medium (FertiPro, Beernem, Belgium), at a volume–volume ratio. It was centrifuged again at 816× *g* for 10 min. The supernatant was removed, and the pellet was resuspended with 0.8 mL of Sequential Fert^TM^ [34].

The last sperm selection technique employed was the Swim-up. This method effectively segregated spermatozoa based on their inherent motility. Initially, an equal volume of culture medium was added to the semen sample. The mixture was then centrifuged at 816× *g* for 10 min, resulting in the formation of an agglomerate of cells. The supernatant was discarded, and 0.8 mL of Sequential Fert^TM^ was then added. After an incubation period of 45 min, a volume of 0.8 mL was carefully collected [35,36]. All sperm selection techniques are summarised in Appendix A.

After subjecting the samples to sperm selection techniques, the resulting purified fraction underwent comprehensive analysis to assess sperm quality parameters, following the guidelines outlined in the WHO 2010 manual. The parameters investigated encompassed concentration, progressive motility, vitality, morphology, chromatin structure stability, and sperm DNA fragmentation. For the analysis of concentration and progressive motility, the Integrated Semen Analysis System (ISAS) developed by PROISER was employed to objectify the results obtained for the parameters. This system utilizes Computer Assisted Semen Analysis (CASA), which combines a phase contrast microscope with specialized software to automate the analysis of the sample. To evaluate vitality, the Sperm VitalStainTM kit (NidaCon International AB, Mölndal, Sweden) was employed. Morphology analysis was carried out using the Bio-DiffTM stain (Biognost, Zagreb, Croatia) and adhered to the Kruger criteria. The stability of the chromatin structure was assessed using aniline blue staining (Sigma-Aldrich, Merck, Darmstadt, Germany). Furthermore, sperm DNA fragmentation was analyzed using the Terminal deoxynucleotidyl transferase dUTP Nick End Labelling (TUNEL) method. In this method, DNA breaks were labelled with FITC-labeled dUTP and subsequently analyzed using flow cytometry. The BD Biosciences’s APO-DIRECTTM kit (BD Pharmigen, BD Biosciences, San Diego, CA, USA) was used for this purpose. Initially, the samples were fixed with 1% paraformaldehyde for 60 min, then washed twice with phosphate-buffered saline (PBS, Gibco, Thermofisher, UK), and stored in 70% ethanol at −20 °C to permeabilize the sperm membrane. To perform the analysis, the ethanol was removed from the samples using a washing solution, followed by incubation with a DNA labelling solution (FITC-dUTP) at 37 °C for 60 min. Subsequently, the samples were rinsed with the washing solution. Samples were incubated with propidium iodide and RNase for 30 min. Negative controls were also prepared, excluding the TdT transferase from the DNA labelling solution to prevent binding of the labelled dUTP to DNA breaks. Finally, using the flow cytometer (CytoFLEX S, Beckman Coulter, Life Science, Indianapolis, IN, USA), the DNA fragmentation of 20,000 spermatozoa was analyzed (Figure 2).

### 2.3. Statistical Analysis

The statistical analysis and interpretation of the obtained results were carried out utilizing the Statistical Package for the Social Sciences software (IBM Corp. Released 2017. IBM SPSS Statistics for Windows, Version 25.0. Armonk, NY, USA). Before analysis, the distribution of variables was assessed using the Kolmogorov–Smirnov test, revealing a departure from normality. Similarly, the Levene test indicated the absence of homoscedasticity among the variables. Consequently, to evaluate the statistical differences between the outcome variables, the Kruskall–Wallis non-parametric test was employed. Results were considered statistically significant when the *p*-value was below 0.05.

## 3. Results

A cohort of 200 patients were recruited for the study with a mean age of 33.7 ± 9.3 years. In the initial set of 100 patients, after conducting an evaluation of sperm quality on the fresh sample and reserving an aliquot for subsequent DNA fragmentation analysis, the remaining sample was equitably partitioned into two identical volumes. One portion underwent processing through DGC, while the other was processed using the microfluidic device (SwimCount^TM^ Harvester). The second set of 100 patients was designated for the comparative analysis between the Swim-up technique and the microfluidic device. This comparison was carried out employing the same methodology as that applied to the initial 100 patients.

### 3.1. Relationship between DNA Fragmentation and Sperm Quality Parameters

Out of the 200 patients, sperm DNA fragmentation analysis was conducted on 181 unprocessed samples. These samples were subsequently stratified into quartiles based on their DNA fragmentation levels. Notably, our analysis revealed a statistical relationship between DNA fragmentation and parameters such as progressive sperm motility, vitality, and sperm chromatin stability values, as presented in Appendix A.

### 3.2. Comparison between the Microfluidic Device and Density Gradients

In the initial phase of the study, when the samples were processed with the SwimCount^TM^ Harvester, an increase in sperm concentration was observed in comparison to the DGC group. However, this increment did not attain statistical significance (8.05 × 10^6^/mL (IQR: 3.55–15.23) vs. 7.15 × 10^6^/mL (IQR: 3.35–15.45); *p* = 1.00). On the contrary, progressive sperm motility showed a significant increase when the microfluidic device was used in comparison with DGC. (80.00% (IQR: 75.00–84.00) vs. 75.00% (IQR: 69.00–79.00); *p* = 0.003). The analysis of the total progressive motile sperm counts also revealed a significant increase when comparing the device to density gradient methods. (4.71 × 10^6^ (IQR: 2.12–10.07) vs. 2.69 × 10^6^ (IQR: 1.39–6.13); *p* = 0.047). Moreover, a significant increase in sperm vitality was observed in the SwimCount^TM^ Harvester group (89.00% (IQR: 85.00–91.00) vs. 80.00% (IQR: 74.00–86.00); *p* < 0.001). Analogously, there was a significant improvement in sperm morphology when the microfluidic device was employed (4.00% (IQR: 3.00–5.00) vs. 3.00% (IQR: 2.25–4.00); *p* = 0.039). Using the microfluidic device, a positive tendency was observed for the stability of the chromatin structure. Nevertheless, this difference was not significant (79.00% (IQR: 75.00–83.25) vs. 76.00% (IQR: 70.00–82.00); *p* = 0.149). The last sperm quality parameter studied was DNA fragmentation. In this specific measurement, a significant decrease was observed when samples were processed using SwimCount^TM^ Harvester compared to DGC. (4.14% (IQR: 2.07–9.01) vs. 12.14% (IQR: 6.72–22.44); *p* < 0.001). All these results are summarized in Table 1.

### 3.3. Stratification of the Data by Comparing the Microfluidic Device with Density Gradients

Afterward, data were stratified by quartiles of the most relevant sperm quality parameters. Notably, non-statistically significant differences were found between sperm concentration quartiles. Conversely, when parameters such as progressive motility and the total motile sperm counts were analyzed, statistically significant differences were identified between DGC and the microfluidic device in all quartiles, except for the first quartile. Furthermore, significant differences in DNA fragmentation were observed across all quartiles between both techniques, as illustrated in Figure 3 and Appendix A.

### 3.4. Comparison between the Microfluidic Device and Swim-Up

In the subsequent phase of our study, a comparative analysis between the microfluidic device and the Swim-up selection technique was conducted. The SwimCount^TM^ Harvester group exhibited a significant increase in sperm concentration 14.00 × 10^6^/mL (IQR: 8.70–25.00) compared to the Swim-up group 8.95 × 10^6^/mL; (IQR: 5.20–15.08) *p* = 0.002. A similar trend was observed for total progressive motile sperm count (9.69 × 10^6^; IQR: 6.16–15.91 vs. 6.05 × 10^6^; IQR: 3.44–10.34; *p* = 0.002) and sperm vitality (96.00%; IQR: 93.00–98.00 vs. 94.00%; IQR: 91.00–97.00; *p* = 0.038). Conversely, no statistically significant improvement was observed in the case of progressive sperm motility (89.00%; IQR: 85.00–91.00 vs. 87.00%; IQR: 84.00–91.00; *p* = 1.00), sperm morphology (5.00%; IQR: 4.00–6.00 vs. 4.00%; IQR: 3.00–5.00; *p* = 0.05), stability of chromatin structure (90.50%; IQR: 89.00–92.75 vs. 90.00%; IQR: 86.50–92.00; *p* = 0.487), and sperm DNA fragmentation (3.13%; IQR: 1.62–5.52 vs. 4.17%; IQR: 1.74–8.13; *p* = 0.161), as presented in Table 2.

### 3.5. Stratification of Data Comparing the Microfluidic Device with Swim-Up

Along similar lines, upon segmenting the sperm quality parameter data into quartiles, differences were detected between the SwimCount^TM^ Harvester and Swim-up techniques in the different quartiles. Specifically, when considering concentration parameters and total progressive motile sperm counts, statistically significant differences were evident for all quartiles. However, in the case of progressive sperm motility, we observed significant differences exclusively in the second quartile. The analysis of DNA fragmentation yielded similar results, with significant differences only in the third quartile, as detailed in Figure 4 and Appendix A.

## 4. Discussion

As has been demonstrated in numerous research studies, the quality of the purified sperms in ART is of vital importance, given its direct impact on success rates and the risk of birth defects [37,38,39]. Currently, the most widely adopted methods in clinical practice for sperm selection are DGC and Swim-up. Multiple studies have asserted the similarity in results between these techniques [40,41]. The main objective of our current study is to functionally validate a novel sperm selection method, known as the SwimCount^TM^ Harvester, based on microfluidics. This validation entails a comprehensive comparison with the conventional sperm selection techniques.

Concentration is one of the main parameters analyzed in a semen quality analysis. In our comparative analysis, a non-significant increase was observed when the SwimCount^TM^ Harvester was compared with DGC. Nevertheless, other studies observed a significant decrease in concentration when microfluidic devices were compared to DGC [23,42,43]. It is worth mentioning the inherent disparities in these outcomes, which can be attributed to the different microfluidic devices employed across the various studies. Otherwise, in comparison of the SwimCount^TM^ Harvester with the Swim-up, sperm concentration increased significantly when the microfluidic device was used. The findings of our research align with another study, which similarly documented an increase in sperm concentration [21].

In the context of progressive sperm motility, it is well-established that a higher percentage of progressively motile sperm is associated with improved reproductive outcomes [44,45,46]. Our study highlights that the SwimCount^TM^ Harvester device significantly enhances the progressive sperm motility fraction after purification in comparison to DGC. The same results were obtained in other studies [23,42,43,47]. On the other hand, when SwimCount^TM^ Harvester was compared with Swim-up, a non-significant increase was observed. However, other studies have found statistically significant improvements [33,43,48,49]. The observed discrepancy may arise due to the positive correlation between the restrictiveness of a sperm selection device and the resulting elevated percentage of progressive spermatozoa. However, a concomitant decrease in sperm concentration is consistently documented across these studies. A delicate balance must be struck to achieve a maximum percentage of progressive motile sperm while mitigating the decrease in sperm concentration. This equilibrium is essential for the applicability of such methods across diverse sample types, encompassing even oligozoospermic samples. Therefore, further studies comparing different microfluidic devices are needed to determine which device might benefit each type of patient based on their sample.

Sperm concentration and motility are inherently interconnected; higher sperm concentration can compensate for reduced progressive motility, and conversely [50]. Our results show that the use of the SwimCount^TM^ Harvester resulted in a significant increase in total progressive motile sperm count compared to the DGC and Swim-up techniques. Similar results were obtained in other studies showing an increase in the total progressive motile sperm count for the microfluidic group [23,43,51]. These findings highlight a remarkable enhancement in the sperm parameter widely regarded as the foremost predictor of reproductive success [52,53].

After categorizing the outcomes of concentration, progressive motility, and total progressive motile sperm count into quartiles, no significant differences are noted across all quartiles. Nevertheless, higher results are observed for the SwimCount^TM^ Harvester group, except for the last quartile of the progressive motility parameter. These results, both overall and stratified by quartiles, underscore the versatility of the SwimCount^TM^ Harvester device. This suggests that the applicability of this device extends seamlessly to scenarios where samples would be processed using conventional sperm selection techniques. Furthermore, these results confirm the good results observed for the microfluidic group compared to conventional techniques [21,23,43,47,48,51]. The observed non-statistical improvements may be caused by the low sample size after the stratification.

Closely related to progressive motility is sperm vitality [54,55]. It is essential to recognize that sperm vitality plays a pivotal role in reproductive outcomes [56,57]. Additionally, prior studies have demonstrated the efficacy of microfluidic devices in significantly enhancing the selection of live sperm [33,51]. Consistent with these findings, our study revealed that the SwimCount^TM^ Harvester device exhibited a significant increase in the percentage of live spermatozoa compared to the two conventional selection methods. The reason for this increase could be due to the minimization of sample handling, as well as avoiding the mechanical stress of centrifuging the samples. These results affirm that the device allows a higher proportion of live sperm to be selected, thus indicating that the materials used in the device do not have adverse effects on cell viability.

Spermatogenesis entails a series of morphological transformations that convert a spermatid into a mature spermatozoon [58]. These transformations are not entirely uniform, leading to a notable proportion of spermatozoa exhibiting various morphological abnormalities [59]. Nevertheless, the role of sperm morphology as a predictor of reproductive success remains uncertain [60,61,62,63]. In our research, the SwimCount^TM^ Harvester device demonstrated an increase in the proportion of morphologically normal spermatozoa. This increase was significant when SwimCount^TM^ Harvester was compared to DGC and not significant when compared to Swim-up. Although there was a trend towards better morphology when the microfluidic device was used. The rise in the percentage of morphologically normal spermatozoa may be attributed to the microfluidic device’s membrane, which closely matches the size of a morphologically normal spermatozoon. Consequently, this similarity could lead to an increase in the proportion of normal spermatozoa captured within the device. Furthermore, similar results were observed in a different microfluidic device, where the percentage of spermatozoa with normal morphology was also higher when compared to DGC [48] and Swim-up [64].

In the context of genetic information, a spermatozoon adeptly condenses its DNA. This level of genomic compaction is achieved through the replacement of sperm histones with protamines [65,66]. Aberrations in chromatin structure have been demonstrated to result in flawed DNA compaction, potentially leading to sperm DNA damage [67,68]. On a reproductive level, reduced sperm maturation has been significantly linked to worse outcomes [69,70,71,72]. However, some authors argue against a direct association between sperm chromatin stability and reproductive outcomes [73,74]. However, in our results, no significant differences were observed for this parameter.

During spermatogenesis, the sperm cell sheds its cytoplasmic contents, making the sperm nucleus susceptible to the potential deleterious effects of ROS [75], inducing DNA damage [76,77]. Sperm DNA fragmentation has been strongly linked to a significant decrease in the success rates of reproductive treatments [14,69,70]. However, other authors have not seen this association [78]. In our current research, we observed a non-significant increase in sperm DNA fragmentation, with values rising from 10.36% in fresh samples to 12.14% for the DGC group. On the contrary, a significant decrease to 4.14% was observed for the microfluidic group. This outcome is intriguing, particularly since the application of DGC is selecting the optimal sperm subpopulation, and a reduction in DNA fragmentation would be expected. Nevertheless, prior studies have described a similar increase in fragmentation when DGC was used [79,80,81]. It is noteworthy that centrifugation has been shown to elevate ROS production, inducing oxidative stress [82]. Furthermore, the culture media employed in the preparation of density gradients contain transition metals like iron and copper, which exhibit an affinity for nucleic acids, contributing to DNA fragmentation [7]. In contrast, when comparing the performance of the microfluidic device with the Swim-up technique, both selection methods demonstrated a significant reduction in DNA fragmentation compared to fresh samples. Moreover, DNA fragmentation is not significantly lower for the microfluidic group compared to the Swim-up group. Consistent with our findings, previous studies have reported similar outcomes, illustrating a reduction in sperm DNA fragmentation when the microfluidic device was employed compared to other selection methods, including DGC [21,42,51] and Swim-up [48,49,51]. When stratifying the DNA fragmentation results into quartiles, statistically significant differences were evident across all quartiles in the comparison between SwimCount^TM^ Harvester and DGC. In contrast, when the microfluidic device was compared with Swim-up, significant differences only were observed for the third quartile. It is noteworthy that, despite these differences, the percentage of fragmentation is lower for the microfluidic group, except in the first quartile. Although the implications of using sperm with fragmented DNA in ART are not fully elucidated and remain the subject of debate, microfluidic technologies hold promise as a potential solution in cases where high sperm DNA fragmentation causes embryo development blockages, implantation failures, or pregnancy losses.

In recent years, we have witnessed the emergence of promising novel approaches in the field of microfluidics. This progress is underpinned by reducing waiting times and reproducing the natural process of sperm selection in vivo [21,23,83]. Furthermore, microfluidics has the immense potential of integrating various laboratory procedures into a single automated process [83]. Such integration not only minimizes gamete manipulation but also reduces the stress associated with such manipulation. Moreover, it obviates human error and variability between clinics [28].

A limitation of the study was the lack of analysis of the SwimCount^TM^ Harvester device in frozen sperm samples that would allow us to show whether the microfluidic device is equally effective in this type of sample, which is commonly used in andrology laboratories to carry out assisted reproduction treatments. In addition, the sample size should be increased, especially in pathological sperm samples such as oligoasthenoteratozoospermia, necrozoospermia, or high DNA fragmentation. Although the results were separated by quartiles, the sample size of each quartile was small.

## 5. Conclusions

In conclusion, comparing the microfluidic SwimCount™ Harvester device with DGC showed statistically improved results concerning progressive motile sperm cells, total progressive motile sperm count, vitality, normal morphology, and DNA fragmentation index. Comparing the microfluidic device with the Swim-up techniques, statistically significant enhancements in purified sperm concentration, total progressive motile sperm count, and vitality were discerned. Moreover, the integration of microfluidics affords a reduction in necessary media consumption, laboratory equipment, waiting durations, and overall workload. Furthermore, this approach reduces the risk of human errors, diminishes the need for supplementary sample handling, and simplifies the process of sperm selection. Therefore, as this is a novel technique, further prospective studies are needed to confirm the results obtained in the present study.

## Figures and Tables

**Figure 1 biomedicines-12-01131-f001:**
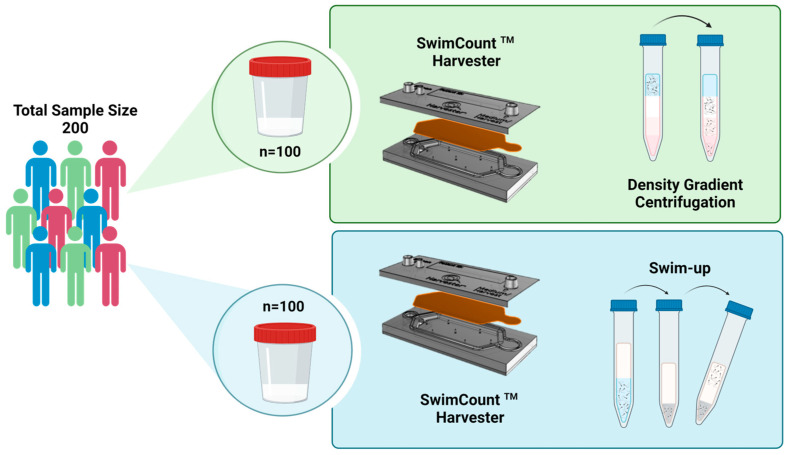
Flow diagram which explains the study design. A total of 200 people were recruited; samples from the first 100 patients were used to compare the SwimCount^TM^ Harvester device with the density gradients and the other 100 different samples were used to compare the SwimCount Harvester with the Swim-up techniques. In each part of the study, the samples were divided into two equal volumes. Created with BioRender.com.

**Figure 2 biomedicines-12-01131-f002:**
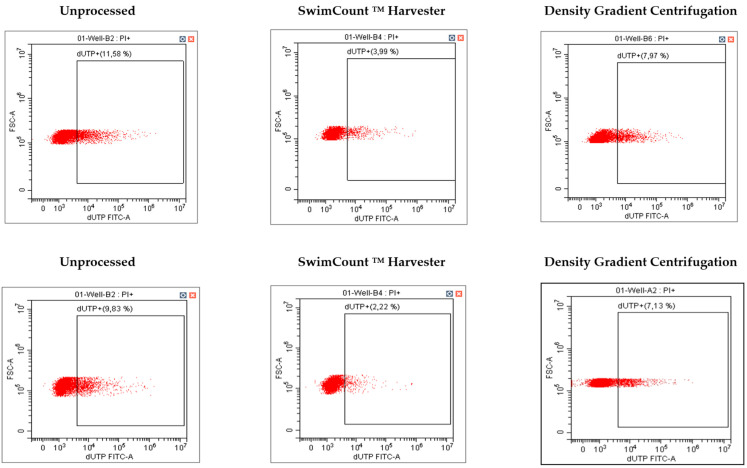
The following graphs represent the results obtained by flow cytometry analysis to assess sperm DNA fragmentation. Each graph represents a different sample processing method.

**Figure 3 biomedicines-12-01131-f003:**
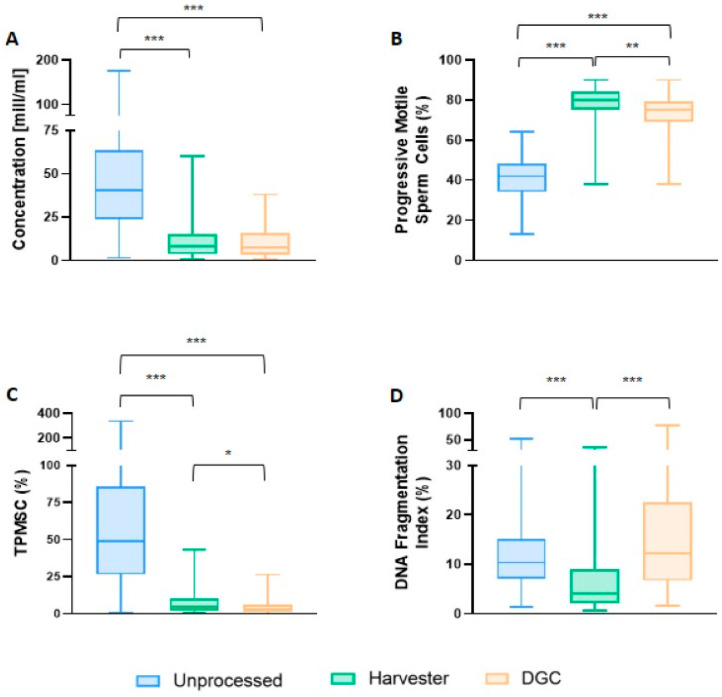
Box plots for each sperm processing method: unprocessed, the microfluidic device, and DGC. Graph (**A**) shows the differences in concentration between the different sperm selection techniques. Graph (**B**) shows the differences in the percentage of progressive motile sperm. Graph (**C**) focuses on the total number of progressive motile sperm count and graph (**D**) on the percentage of DNA fragmentation. The median of each sperm quality parameter is shown as a horizontal value within the shaded box, 25th and 75th percentiles as lower and upper bounds of shaded box, and whiskers demonstrate 1.5 times the upper or lower quartile. * *p* < 0.05; ** *p* < 0.01; *** *p* < 0.0001.

**Figure 4 biomedicines-12-01131-f004:**
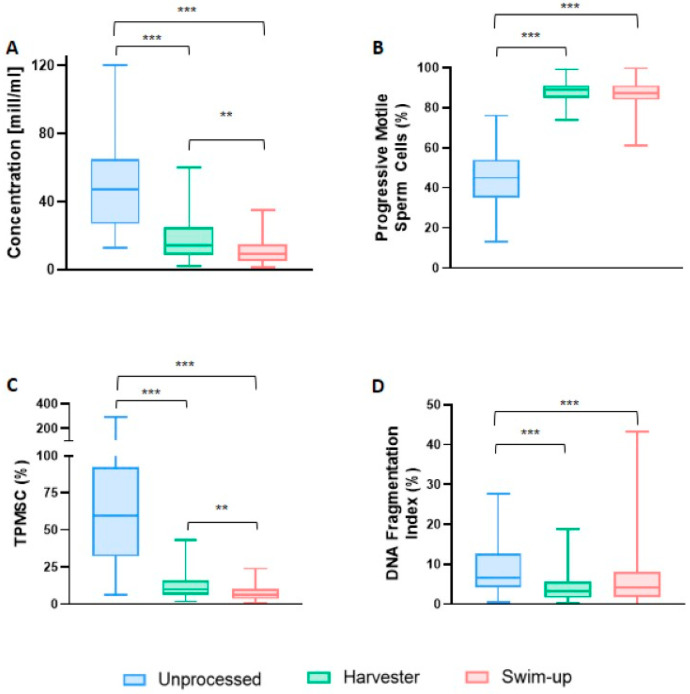
Box plots for each sperm processing method: unprocessed, the microfluidic device, and Swim-up. Graph (**A**) shows the differences in concentration between the different sperm selection techniques. Graph (**B**) shows the differences in the percentage of progressive motile sperm. Graph (**C**) focuses on the total number of progressive motile sperm count and graph (**D**) on the percentage of DNA fragmentation. The median of each sperm quality parameter is shown as a horizontal value within the shaded box, 25th and 75th percentiles as lower and upper bounds of shaded box, and whiskers demonstrate 1.5 times the upper or lower quartile. ** *p* < 0.01; *** *p* < 0.0001.

**Table 1 biomedicines-12-01131-t001:** SwimCount™ Harvester vs. Density Gradient Centrifugation (DGC).

Sperm Quality Parameters Sample Size	Unprocessed	SwimCount™ Harvester	DGC	*p*-Value a	*p*-Value b	*p*-Value c
Sperm Concentration (mill/mL) Median (IQR) *n* =100	40.45 (23.68–63.43)	8.05 (3.55–15.23)	7.15 (3.35–15.45)	*p* < 0.001	*p* < 0.001	*p* = 1.00
Progressive Motile Sperm Cells (%) Median (IQR) *n* = 100	42.00 (34.00–48.00)	80.00 (75.00–84.00)	75.00 (69.00–79.00)	*p* < 0.001	*p* < 0.001	*p* = 0.003
Total Progressive Motile Sperm Count (mill) Median (IQR) *n* = 100	48.77 (26.29–85.50)	4.71 (2.12–10.07)	2.69 (1.39–6.13)	*p* < 0.001	*p* < 0.001	*p* = 0.047
Sperm Vitality (%) Median (IQR) *n* = 78	76.00 (71.00–80.00)	89.00 (85.00–91.00)	80.00 (74.00–86.00)	*p* < 0.001	*p* = 0.009	*p* < 0.001
Normal Morphology (%) Median (IQR) *n* = 88	2.00 (1.00–3.00)	4.00 (3.00–5.00)	3.00 (2.25–4.00)	*p* < 0.001	*p* < 0.001	*p* = 0.039
Sperm Chromatin Stability (%) Median (IQR) *n* = 86	69.00 (64.00–74.00)	79.00 (75.00–83.25)	76.00 (70.00–82.00)	*p* < 0.001	*p* < 0.001	*p* = 0.149
DNA Fragmentation Index (%) Median (IQR) DGC *n* = 90	10.36 (7.12–15.10)	4.14 (2.07–9.01)	12.14 (6.72–22.44)	*p* < 0.001	*p* = 0.583	*p* < 0.001

Semen quality parameters by sperm selection method, the microfluidic device and DGC. (a) Statistically significant differences between unprocessed samples and samples processed by SwimCount™ Harvester. (b) Statistically significant differences between unprocessed samples and samples processed by DGC. (c) Statistically significant difference between SwimCount™ Harvester and DGC.

**Table 2 biomedicines-12-01131-t002:** SwimCount™ Harvester vs. Swim-up.

Sperm Quality Parameters Sample Size	Unprocessed	SwimCount™ Harvester	Swim-Up	*p*-Value a	*p*-Value b	*p*-Value c
Sperm Concentration (mill/mL) Median (IQR) *n* =100	47.00 (26.85–64.80)	14.00 (8.70–25.00)	8.95 (5.20–15.08)	*p* < 0.001	*p* < 0.001	*p* = 0.002
Progressive Motile Sperm Cells (%) Median (IQR) *n* = 100	45.00 (35.00–54.00)	89.00 (85.00–91.00)	87.00 (84.00–91.00)	*p* < 0.001	*p* < 0.001	*p* = 1.00
Total Progressive Motile Sperm Count (mill) Median (IQR) *n* = 100	59.63 (32.06–92.21)	9.69 (6.16–15.91)	6.05 (3.44–10.34)	*p* < 0.00	*p* < 0.001	*p* = 0.002
Sperm Vitality (%) Median (IQR) *n* = 94	85.00 (81.00–88.00)	96.00 (93.00–98.00)	94.00 (91.00–97.00)	*p* < 0.001	*p* < 0.001	*p* = 0.038
Normal Morphology (%) Median (IQR) *n* = 99	2.00 (1.00–3.00)	5.00 (4.00–6.00)	4.00 (3.00–5.00)	*p* < 0.001	*p* < 0.001	*p* = 0.05
Sperm Chromatin Stability (%) Median (IQR) *n* = 92	82.00 (78.00–88.00)	90.50 (89.00–92.75)	90.00 (86.50–92.00)	*p* < 0.001	*p* < 0.001	*p* = 0.487
DNA Fragmentation Index (%) Median (IQR) *n* = 90	6.64 (4.19–12.62)	3.13 (1.62–5.52)	4.17 (1.74–8.13)	*p* < 0.001	*p* < 0.001	*p* = 0.161

Semen quality parameters by sperm selection method, the microfluidic device and Swim-up. (a) Statistically significant differences between unprocessed samples and samples processed by SwimCount™ Harvester. (b) Statistically significant differences between unprocessed samples and samples processed by Swim-up. (c) Statistically significant difference between SwimCount™ Harvester and Swim-up.

## Data Availability

Data contained within the article.

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
