# Peer review of "Can Microfluidics Improve Sperm Quality? A Prospective Functional Study"

_biomedicines, 2024, doi:10.3390/biomedicines12051131_

Round 1
Reviewer 1 Report
Comments and Suggestions for Authors
This clinical study demonstrates the usefulness of the sperm sorting method using a microfluidic device compared to the swim-up method and the density gradient centrifugation method. The number of cases is reasonably sufficient (100 cases each), and the statistical analysis method is also satisfactory, so the results are sufficiently reliable. The discussion was also appropriate, and no particular problems were found.
It would be wonderful if the fertility of the sperm could also be examined, but it may be difficult to collect such a large sample from infertility clinic patients alone.
I have noted a few minor points of concern, please correct them.
1) The word "microfluidics" appears frequently throughout the paper, especially in the latter half of Introduction. This word refers to a system, but in many places it is used to describe a sperm screening method, such as the DGC method or the Swim-up method. Since it is the device that performs the sperm sorting, I think the term "microfluidic device" should be used in many places.
2) page 7 line 230 The section number should be 3.5.
Reviewer 2 Report
Comments and Suggestions for Authors
There are still some defects in this work:
1. Abstract was not well organized and should be re-written. The paper only gave statistic significant analysis, that was not enough for scientific paper and key research data should be supplemented.
2. As we are not very familiar with the mechanism and protocol for microfluidic device, it is wise to introduce the method with figures and simplified illustration.
3. Because microfluidics is still in its early stages of development, any conclusions should be careful.
4. Due to the inherent disparities in these outcomes, which can be attributed to the different microfluidic devices employed across the various studies, much research is needed in standard management of related devices protocols. That can be explained in discussion part.
All in all, this work is well done, and the method seems of some value in selecting sperm in ART process.
Comments on the Quality of English LanguageEnglish is ok, and minor editing is enough.
